# Antimicrobial Dosing During Continuous Venovenous Hemodiafiltration in Septic Shock Patients: A Prospective, Multicenter Study Protocol

**DOI:** 10.3390/antibiotics14040420

**Published:** 2025-04-21

**Authors:** Alicia Wendy Vega Harwood, Marta Martín Fernández, Carlos Ezquer Garin, Francisco Javier Álvarez, Rocío López Herrero, Eduardo Tamayo, Gerardo Aguilar

**Affiliations:** 1Critical Care Unit, Anesthesiology and Critical Care Department, Clinic University Hospital of Valladolid, 47003 Valladolid, Spain; awvega@saludcastillayleon.es (A.W.V.H.); eduardo.tamayo@uva.es (E.T.); 2BioCritic, Group for Biomedical Research in Critical Care Medicine, 47003 Valladolid, Spain; marta.martin.fernandez@uva.es (M.M.F.); alvarez@uva.es (F.J.Á.); 3Personalizing Antimicrobials in Critical Care Unit (PACCU) Network, 46010 Valencia, Spain; cezquer@incliva.es; 4Pharmacology, Faculty of Medicine, University of Valladolid, 47005 Valladolid, Spain; 5Center for Biomedical Research Network on Infection Diseases (CIBERINFEC), Instituto de Salud Carlos III, 28029 Madrid, Spain; 6Institute for Health Research (INCLIVA), Clinic University Hospital of Valencia, 46010 Valencia, Spain; 7Central Unit for Medical Research of the School of Medicine (UCIM), University of Valencia, 46010 Valencia, Spain; 8Department of Pharmacy, Clinic University Hospital of Valencia, 46010 Valencia, Spain; 9Department of Surgery, University of Valladolid, 47003 Valladolid, Spain; 10Critical Care Unit, Anesthesiology and Critical Care Department, Clinic University Hospital of Valencia, 46010 Valencia, Spain; 11Department of Surgery, School of Medicine, University of Valencia, 46010 Valencia, Spain

**Keywords:** continuous renal replacement therapy, acute kidney injury, renal failure, pharmacokinetic, beta-lactams

## Abstract

**Background:** Sepsis is a major global health issue and the leading cause of death in critically ill patients, with rising incidence and associated healthcare costs. Early administration of antibiotic therapy is crucial, but increasing antibiotic resistance poses a threat. Beta-lactam antibiotics, commonly used as a first-line therapy option against sepsis, often demonstrate unpredictable concentrations due to pharmacokinetic and pharmacodynamic changes in critically ill patients. Acute kidney injury (AKI) affects a significant portion of septic patients, and continuous renal replacement therapy can further complicate treatment by reducing antibiotic levels and, consequently, increasing antibiotic resistance risk. **Objectives:** To develop pharmacokinetic/pharmacodynamic models for beta-lactam antibiotics in septic shock patients undergoing continuous renal replacement therapy (CRRT), with the goal of optimizing antibiotic dosing and then improving treatment outcomes. **Methods:** Septic shock Caucasian adult patients treated with beta-lactams and who have undergone major surgery in AKI failure that requires CRRT will be eligible with previous informed written consent. CRRT will be performed exclusively using Continuous Venovenous Hemodiafiltration (CVVHDF) modality. Antimicrobial determination analyses will be carried out with LC-MS/MS. Further calculation of pharmacokinetic parameters and determination of PK/PD breakpoints will be made using Monte Carlo simulation. **Conclusions:** The expected results from this study will lead to a better understanding of the pharmacokinetics of beta-lactam antibiotics in critically ill patients with AKI and septic shock undergoing CVVHDF, allowing for improved therapeutic strategies.

## 1. Introduction

Sepsis is a major global health concern that affects around 25 million people worldwide. It is the principal cause of death in critically ill patients, potentially leading to 6 million deaths [1,2]. While classically defined in 1991 as the systemic inflammatory response syndrome (SIRS) triggered by infection, our understanding of sepsis has since evolved. In 2016, sepsis was redefined as a life-threatening organ dysfunction caused by the host’s dysregulated response to infection. Septic shock is defined as sepsis with persistent hypotension requiring vasopressors to maintain a mean arterial pressure above 65 mmHg and lactate levels exceeding 2 mmol/L despite adequate fluid resuscitation [3,4].

The clinical and economic burden of sepsis is substantial and growing, with increasing incidence and sepsis-related deaths. It affects approximately 1 out of 3 intensive care unit (ICU) patients, contributing to elevated healthcare costs due to prolonged hospital stays and associated morbidity. This has led the World Health Organization (WHO) to recognize sepsis as a global health priority [1,5,6,7,8,9,10,11,12,13,14]. The incidence of sepsis demonstrates significant variation across different regions and populations. For instance, Germany reported an increase in the adjusted in-hospital incidence from 256 to 335 cases per 100,000 persons per year between 2007 and 2013 [7]. Similarly, in Australia, sepsis-coded hospitalizations increased from 36,628 in 2002–2003 to 131,826 in 2020–2021 [12]. In Spain, our research group’s study revealed a rise in incidence from 200 cases per 100,000 inhabitants in the year 2000 to 480 cases per 100,000 in 2013, with a mortality rate of 56 cases per 100,000 inhabitants in 2000 and 830 cases per 100,000 inhabitants in 2013 [15]. This means that each year in Spain, there are around 50,000 cases of severe sepsis, of which 17,000 result in death [16]—figures of incidence and mortality that exceed those of other diseases with significant social impact, such as breast cancer or AIDS [17].

Early identification of causative pathogens and the prompt administration of targeted antibiotic treatment are essential for improving prognosis. However, the alarming increase in global antibiotic resistance may compromise this approach in the future. The need to combat resistance necessitates a focus on avoiding the inappropriate use of antibiotics and optimizing their prescription to ensure effective treatment while minimizing the selection of resistant strains [18,19,20]. Beta-lactam antibiotics are commonly employed as the first-line treatment for patients with sepsis or septic shock due to their broad spectrum of activity and generally favorable safety profile. These antibiotics, which include penicillins, cephalosporins, carbapenems, and monobactams, inhibit bacterial cell wall synthesis, leading to bacterial death. However, these hydrophilic antibiotics are particularly prone to variability in plasma concentrations in critically ill patients due to physiological alterations such as increased vascular permeability, which leads to a higher distribution volume and greater intra- and inter-individual pharmacokinetic/pharmacodynamic (PK/PD) variability [19,21]. This variability complicates dosing and increases the risk of treatment failure.

Acute kidney injury (AKI) is a frequent complication in septic patients, occurring in 40–50% of cases and further impacting treatment strategies [22]. The incidence is even higher in septic shock, with 64% developing AKI and 9–20% of these requiring continuous renal replacement therapy (CRRT) [23,24]. Continuous hemodiafiltration (CHDF), a widely used CRRT modality in intensive care units, combines the principles of hemodialysis and hemofiltration and can impact pharmacokinetics by promoting extracorporeal drug elimination. This can result in lower antibiotic blood concentrations, thereby increasing the risk of treatment failure due to subtherapeutic levels and potentially contributing to the emergence of antibiotic resistance, creating a critical challenge in antibiotic management [25,26,27].

Therefore, one of the difficulties in treating septic patients, particularly those undergoing CRRT, is the need for individualized dose adjustment to account for the substantial variability in antibiotic pharmacokinetics. This variability arises from a complex interplay of factors, including physiological changes, the aforementioned high PK/PD variability, the specific CRRT modality employed, the type of filter membrane used, and various treatment parameters [28]. Consequently, clinical recommendations for patients with normal renal function do not apply to renal patients undergoing CRRT [20]. In this context, developing PK/PD models for beta-lactam antibiotics in septic shock patients receiving CRRT could provide valuable tools to assess drug exposure and guide optimal dosage adjustments, ultimately improving treatment outcomes.

Previous studies have reported high variability in antibiotic exposure and the achievement of therapeutic targets in patients receiving CRRT, highlighting the need for individualized dosing strategies [20,25,26]. The most effective strategy to optimize antimicrobial therapy is monitoring plasma antibiotic levels to ensure therapeutic efficacy, which has become increasingly important in ICUs over the past decades [20,29,30,31]. In fact, the latest Surviving Sepsis Campaign international guidelines strongly recommend optimizing beta-lactams through a PK/PD-based approach to improve patient outcomes [20]. However, many hospitals lack the necessary technology to monitor the plasma levels of most antibiotics.

This project, therefore, represents a pioneering study to evaluate beta-lactam antibiotics PK/PD in critically ill septic shock patients undergoing CRRT. The primary goal is to generate pharmacokinetic data that will enable the optimization of antibiotic dosing in this challenging patient population, ultimately contributing to improved treatment efficacy and a reduction in the development of antibiotic resistance.

## 2. Experimental Design

### 2.1. Type of Study

We present a prospective, multicenter study that includes 140 adult (>18 years old) patients admitted to the Hospital Clínico Universitario de Valladolid and Hospital Clínico Universitario de Valencia post-surgical ICUs, who have undergone major surgery, with septic shock and AKI that requires CRRT and treated with beta-lactam antibiotics.

### 2.2. Study Population

#### 2.2.1. Eligibility Criteria

To identify patients with septic shock, we will follow SEPSIS-3 criteria which include the presence of both [4]:Confirmed diagnosed infection;Persistent hypotension requiring vasopressors to maintain a mean arterial pressure (MAP) >65 mmHg or greater and having a serum lactate level >2.0 mmol/L (>18 mg/dL) despite adequate volume resuscitation.

Patients with microbiological confirmation of infection through standard bacterial cultures or proteomic (MALDI-TOF) or molecular (SeptiFast, Roche) techniques will be considered, guided by the infection focus.

Patients will be divided into 14 groups of 10 patients each (10 patients per each beta-lactam antibiotic subject of study).

Inclusion criteria
Caucasian race patients over 18 years old, presenting septic shock defined by SEPSIS-3 consensus [4] and requiring CRRT (Continuous Venovenous Hemodiafiltration (CVVHDF) for AKI treatment;Informed consent is given by the patient or family members.
Exclusion criteria
Non-Caucasian race patients;Pregnant women;Patients in agonizing condition;Limitation of therapeutic effort.


#### 2.2.2. Criteria for Initiating CVVHDF [32,33]

Metabolic:
Azotemia (Urea > 100 mg/dL)Uremic complications (uremic encephalopathy, pericarditis, bleeding)Hyperkalaemia (K^+^ > 6 mmol/L and/or ECG changes)Hypermagnesemia (Mg^2+^ > 4 mmol/L and/or anury/absence of tendon reflexes);
Acidaemia (pH ≤ 7.15);Oligury, urine output <200 mL/12 h or 6 h duration anury;Fluid overload (e.g., diuretic resistant organic edema in relation to AKI).

#### 2.2.3. Flow Diagram: Study Population Selection

Patient admitted to ICU at Hospital Clínico Universitario de Valladolid or Hospital Clínico Universitario de Valencia post-major surgery, potentially in septic shock →Does the patient meet SEPSIS-3 criteria?→ |No| Z [Exclude from Study]→ |Yes| C {Microbiological Confirmation of Infection}→ |No| Z→ |Yes| D {Assess Inclusion Criteria}→ |No| Z→ |Yes| E {Verify Exclusion Criteria}→ |Meets ≥ 1| Z→ |Does Not Meet| F [Include in Study]

### 2.3. Variables Studied

At the time of patient inclusion in the study, demographic, clinical, and analytical data (leukocytes, lymphocytes, neutrophils, platelets, bilirubin, creatinine, glucose, C-reactive protein (CRP), lactate dehydrogenase (LDH), procalcitonin, D-dimer, and lactate) will be recorded to describe the clinical phenotype and severity stratification according to the SOFA [34] and APACHE II [35] scales. Follow-up variables will also be recorded, including the duration of CVVHDF therapy, hospital and ICU stay, mechanical ventilation duration, ICU mortality, and 28-day mortality, as well as any complications that may arise.

### 2.4. Statistical Analysis

Statistical analyses will be performed using IBM SPSS Statistics for Windows (Version 22.0). IBM Corp. (Armonk, NY (USA) and R software (R Core Team. (2023). R: A language and environment for statistical computing, version 4.3.2). R Foundation for Statistical Computing. https://www.R-project.org/, accessed on 13 April 2025) by an experienced statistician. Demographic and clinical characteristics will be assessed using the Chi-square test, Likelihood Ratio test, Fisher’s exact test for categorical variables, Student’s *t*-test for independent samples, or the Mann–Whitney U test, as appropriate.

### 2.5. Ethical Considerations

Our protocol was approved by the Research Ethics Committee for Medicinal Products (CEIm) of the Clinic University Hospital of Valladolid (code: PI 23-5-C).

### 2.6. Limitations

We consider the limitations of this study to be the following:

Only surgical sepsis is studied, without considering medical sepsis at any point.

## 3. Materials and Methods

### 3.1. Description of Materials

#### 3.1.1. Continuous Venovenous Hemodiafiltration (CVVHDF)

CRRT in this protocol will be delivered exclusively using CVVHDF. This modality is widely used in post-surgical ICU patients to treat patients with unstable hemodynamic pathologies as well as to remove cytokines in sepsis patients, even if they have normal renal function [36].

Regional citrate anticoagulation (RCA) circuit will be implemented as this modality of anticoagulation prevents filter clotting and lessens the risk of bleeding manifestations compared to other anticoagulation regimens, making it a reasonable alternative, especially in active bleeding, coagulopathy, or thrombocytopenia. When citrate mixes with blood, it binds to ionized calcium, which is fundamental for the coagulation process, preventing the generation of thrombin. The citrate-calcium complex in the blood exiting the filter through the venous line is administered to the patient. In the liver (primarily, but also in the kidneys and skeletal muscles), this complex is metabolized into bicarbonate (1 mmol of citrate is converted into 3 mmol of bicarbonate); therefore, it influences the acid-base balance. When ionized, calcium is released and becomes available as a coagulation factor in the bloodstream. Systemic anticoagulation does not occur if the serum ionized calcium concentration is maintained at a physiological level (1.0–1.3 mmol/L or 4–5.2 mg/dL) [37,38].

##### CVVHDF with Citrate Materials (Prismaflex) [39]

Priming solution: 1 L of saline solution 0.9% NaCl saline without sodium heparin.Anticoagulant: citrate solution bag (Regiocit^®^).This solution is indicated as a replacement fluid for continuous renal replacement therapy (CRRT) with regional citrate anticoagulation.Dialysis solution bag without calcium (Prism0cal^®^).Convection/replacement solution bag without calcium (Hemosol^®^ or 1 L 0.9% saline as per medical indication).Effluent bag.50 mL syringe.5 ampoules of calcium chloride 9.14 meEq.Special line for calcium infusion to the patient.

##### Solution Preparation

The citrate solution (Regiocit^®^) will be placed on the white scale and connected to the white line.For the convection-replacement solution, we will use HEMOSOL^®^ (scale and purple line).Calcium-free solution Prismo0cal^®^ (scale and green line).Effluent bag (scale and yellow line).

##### Calcium Syringe Installation

Connect the calcium special line (which will be pure calcium chloride) without connecting it to the patient and without clamping it. Purge the syringe and line according to the instructions.At the start of the treatment, connect this line to the patient’s central venous catheter (CVC), and if no CVC light is available, place a 3-way valve on the entry light (blue) of the dialysis catheter.

##### Prime the Circuit

Saline solution 0.9% NaCl 0.9%; 1 L will be needed.

##### Flow Settings

See further along.

#### 3.1.2. Liquid Chromatography-Mass Spectrometry (LC-MS/MS)

Liquid Chromatography-Mass Spectrometry analyses is carried out at the Instrumental Techniques Laboratory of the University of Valladolid using a SCIEX QTOF X500R system coupled with a SCIEX EXION LC 2D-UHPLC series (AB Sciex Pte. Ltd., Coppell, TX, USA).

### 3.2. Methods

#### 3.2.1. Clinical Procedure

##### Initial Dosage Flows

Hemodiafiltration protocol total therapy dose by adding ultrafiltration (convection, inferred from the prescribed substitution dose) and diffusion (dialysis) will be between 30–37 mL/kg/h, as shown in Table 1.

##### Percentage of Convection and Diffusion: According to Table 1

The percentage of therapy for each component can be inferred from data obtained in Table 1 and may vary depending on each prescription. The convection dose is obtained by subtracting the dialysis fluid flow from the total therapy dose adjusted to the patient’s weight.

##### Type of Membrane

The type of filter membrane may affect antibiotic clearance and should be considered in critical patients requiring CRRT [40]. Two types will be used, oXiris and ST-150. The oXiris membrane will be applied during the first 48–96 h in surgical patients with septic shock due to its adsorptive capacity to control the cytokine storm [41]. The ST150 membrane is also part of the Prismaflex sets and is made of AN69 ST hollow fiber. This membrane is designed for patients with larger body volumes or higher therapeutic needs in continuous hemofiltration therapy [42]; subsequently, the ST-150 membrane will be employed in patients who have stabilized but still require hemodiafiltration.

This design will allow us to analyze how the adsorption capacity of oXiris impacts the PK/PD of a specific beta-lactam, under identical therapy doses and convection-diffusion ratios, in comparison to the subgroup treated with ST-150, providing valuable insights into the differential effects of adsorption (oXiris) versus standard hemofiltration (ST-150) on the PK/PD behavior of beta-lactams.

Membrane change will be made every 24 h.

##### Sampling Ports

Sampling ports are pre-filter (red), post-filter (blue), diuresis (if not anuric, from the port of the urethral catheter), and ultrafiltrate (yellow).

##### Sample Handling

Blood samples will be collected from the access port of the extracorporeal circuit near the filter based on the pharmacokinetic behavior of each antibiotic (see Table A1). Simultaneously, dialysate filtrate (FD) samples will be collected from the filter tube to determine the sieving coefficient (SC) and CRRT clearance [36]. After collection, each blood sample will be sent to the Hospital Clínico Universitario de Valladolid (HCUV) Biobank and immediately centrifuged. Plasma and supernatants from the FD samples will be immediately frozen at −80 °C and stored until further analysis at the Instrumental Techniques Laboratory of the University of Valladolid.

In parallel, a blood sample will be drawn from each patient into an EDTA tube for hematological, biochemical, and acute phase reactant determinations. The plasma will be separated, aliquoted, and stored under optimal conditions at the same hospital within 2 h of extraction.

##### Sample Preparation for Antimicrobial Determination

Initially, 50 µL of the sample will be extracted with 50 µL of 10% trichloroacetic acid solution and mixed in a vortex for 30 s; then, 100 µL of methanol will be added and mixed again in a vortex.

In some cases, if necessary and according to the protocol, 50 µL of the sample will be extracted by adding 150 µL of methanol. Specifically, 2 µL of the supernatant will be injected into the LC-MS/MS system after centrifugation at 11,200× *g* for 10 min. All solutions and mobile phases will be filtered through 0.45-µm nylon membrane filters purchased from Micron Separations (Micron Separations Inc., Westborough, MA, USA).

##### Quantitative Determination of Antimicrobials

The following beta lactam antibiotics will be evaluated: amoxicillin/clavulanic acid, piperacillin/tazobactam, ampicillin/sulbactam, ceftriaxone, cefotaxime, cefepime, ceftazidime/avibactam, ceftolozane/tazobactam, imipenem/cilastatin, meropenem, ertapenem, meropenem/vaborbactam, cefiderocol, aztreonam-avibactam. These beta-lactam antibiotics have been selected because they are the most widely used and are included in the pharmacotherapeutic guide of the HCUV.

LC-MS/MS analyses will be carried out at a SCIEX QTOF X500R system coupled with a SCIEX EXION LC 2D-UHPLC series (AB Sciex Pte. Ltd., USA). Gradient separation chromatography will be performed using a column with mobile phase A consisting of 0.1% formic acid in water, and mobile phase B of 0.1% formic acid in acetonitrile. The percentage of solvent B will start at 5% for 0.1 min, reach 100% in 2 min, and be maintained for 1.5 min at a flow rate of 500 µL/min. The column will then be reconditioned to 5% B for 1.5 min, for a total run time of 5 min. The column temperature will be maintained at 50 °C. Ionization will be achieved using electrospray ionization (ESI) with a spray voltage of 4000 V for positive mode and 3000 V for negative mode. Nitrogen will be used as both nebulizer and auxiliary gas, set at 50 and 20 arbitrary units, respectively. The vaporizer and ion transfer tube temperatures will both be set at 300 °C. For collision-induced dissociation, high-purity argon will be used at a pressure of 1.5 mTorr. Analytes and internal standards (IS) will be detected using selected reaction monitoring (SRM) of specific transitions.

#### 3.2.2. Pharmacokinetic and Chromatographic Analysis

##### Calculation of Pharmacokinetic Parameters

Non-compartmental and compartmental analyses will be conducted using PK analysis software (Program Abbott Based Pharmacokinetic System PKs, version 1.10, Abbott Laboratories Diagnostic Division (Abbott Laboratories, Illinois, EE.UU.). The area under the concentration-time curve (AUC) for antibiotics will be calculated using the logarithmic trapezoidal rule. Systemic clearance (SC) will be determined as AUC_FD_/AUC_plasma_. The AUC following the first administration will be calculated from 0 h to infinite, and subsequent AUCs will be estimated from 0 h to the beginning of the next infusion interval, although this may vary depending on the half-life of each antibiotic in healthy individuals. Differential equations will be applied to mass balance calculations. These equations will be used to derive formulas for drug concentrations (μg/mL) in the central compartment (C1) and FD (C3) at time t (hours) during and after administration at steady state [36].

Clearance in continuous renal replacement therapy (CRRT clearance, CL CRRT) will be calculated as (QF + QD) − SC, and clearance by non-CRRT pathways (CL non-CRRT) will be estimated as k10 − V1. The concentration-time data for antibiotics in plasma and FD will be simultaneously fitted to the multi-compartmental model described above [36].

##### Calculation of PK/PD Breakpoints for Beta-Lactams Using Monte Carlo Simulation

Monte Carlo simulation will be performed using the random number generation function in Microsoft^®^ Excel^®^ 2016 (Microsoft, Redmond, WA, USA). A set of 10,000 population parameter cases will be generated using the mean and variance of k21, V1, α, and β obtained through the standard two-stage method (compartmental analysis), thereby simulating the plasma concentration transition of antibiotics for 10,000 cases [36].

After setting the dosing interval and administration time for each antibiotic in the aforementioned formula (C1), the exposure time during which the plasma concentration remained at the MIC value (T > MIC) will be calculated as the accumulated percentage over a 24-h period in the plasma concentration transition of each antibiotic for the 10,000 cases. The number of cases showing T > MIC of 40% or more at each target PK/PD value in each regimen will then be calculated. The MIC at which more than 80% of cases achieve this will be established as the probability of reaching the PK/PD breakpoint target for each regimen [36].

## 4. Detail Procedure

We present a prospective, multicenter study protocol that will be conducted at Hospital Clínico Universitario de Valladolid and Hospital Clínico Universitario de Valencia. Our protocol was approved by the CEIm of the Clinic University Hospital of Valladolid (code: PI 23-5-C).

Septic shock Caucasian adult patients treated with beta-lactams and who have undergone major surgery in AKI failure that requires CRRT will be eligible with previous informed written consent. CRRT will be performed exclusively using CVVHDF modality delivered by the PrismaFlex (Baxter) system and with an RCA.

The hemodiafiltration protocol total therapy dose will be between 30–37 mL/kg/h. Dialysate flow rates (Prism0cal solution) range from 1000 to 1800 mL/h depending on the patient’s weight, just like blood flow (see Table 1).

The citrate solution (Regiocit^®^) will be placed on the white scale and connected to the white line as aforementioned. The pre-filter infusion rate of Regiocit should be prescribed and adjusted in relation to the blood flow rate to achieve a target blood citrate concentration of 3 to 4 mmol/L. The extracorporeal circuit anticoagulation flow rate should be adjusted to maintain a post-filter ionized calcium concentration in the range of 0.25 to 0.35 mmol/L. In CVVHDF, the recommended flow rate is 1–2 L/h with a blood flow rate between 100 and 200 mL/min.

Two filter membranes will be used (oXiris and ST-150). In surgical patients with septic shock, the oXiris membrane will be used during the first 48–96 h for its adsorption properties. Once stabilized, patients who still require hemofiltration—particularly those with higher body volume or greater therapeutic needs—will transition to the ST-150 membrane. The oXiris’s adsorption impact on beta-lactam PK/PD will be assessed and compared to ST-150 under identical therapy conditions.

The blood sampling from patients will be carried out by the nursing staff following a detailed procedure for each drug. For this, the instructions specified in Table A1 will be followed. These instructions are based on a set of conditions used in previously published studies. To avoid sample hemodilution, a sufficient volume of blood will be discarded, and subsequently, ~2–3 mL will be collected in the appropriate tube for each analyte.

Plasma extraction from the blood samples will be performed immediately after the sample is taken from the patient, and the instructions detailed in Table A1 for their storage will be followed.

Liquid Chromatography-Mass Spectrometry analyses will be carried out at the Instrumental Techniques Laboratory of the University of Valladolid using a SCIEX QTOF X500R system (SCIEX, Framingham, MA, USA) coupled with a SCIEX EXION LC 2D-UHPLC series (AB Sciex Pte. Ltd., USA). Chromatography will involve a gradient with formic acid in water and acetonitrile as mobile phases. Electrospray ionization (ESI) will be used for detection.

Calculation of Pharmacokinetic Parameters: AUC, systemic clearance (SC), and CRRT clearance (CL CRRT) will be calculated using PK software. Non-compartmental and multi-compartmental modeling will be applied to concentration-time data from plasma and FD.

PK/PD Breakpoints via Monte Carlo simulation: Monte Carlo simulation will generate 10,000 cases to calculate T > MIC (time above the minimum inhibitory concentration) and determine the MIC value where 80% of cases achieve the PK/PD breakpoint target for each antibiotic regimen.

## 5. Expected Results

With this study, we aim to define PK/PD models for each of the aforementioned drugs to optimize the prescription of beta-lactam antibiotics in septic shock patients undergoing CVVHDF, and consequently, avoid subtherapeutic dosing that could lead to antibiotic treatment failure.

The study will provide a detailed understanding of beta-lactam’s PK/PD in critically ill septic shock patients with AKI undergoing CVVHDF. It is expected that the systemic clearance (SC) of these antibiotics will be altered by the dialysis process, particularly considering the use of two different filter membranes (oXiris and ST-150) with varying adsorption properties. A comparative analysis of these two membranes will be conducted to evaluate how the adsorption properties of the oXiris membrane affect the pharmacokinetics of beta-lactam antibiotics compared to the ST-150 membrane. The results are expected to show differences in drug retention or removal between these two filters, which could have implications for optimizing antibiotic dosing in septic shock patients.

The AUC for each antibiotic will be calculated to understand the exposure of patients to these drugs over time and help clarify how dialysis affects the AUC and systemic clearance of antibiotics in this context.

Monte Carlo simulations will be used to calculate the PK/PD breakpoints for beta-lactams in this population, based on the T > MIC (time above minimum inhibitory concentration) for each drug [36]. It is anticipated that these simulations will identify the optimal dosing strategies for beta-lactams in patients receiving CVVHDF.

The results will enable the identification of the MIC values that allow for effective treatment in more than 80% of cases, helping to define appropriate dose adjustments for individual patients based on their specific PK/PD characteristics during CRRT.

The study will calculate the clearance during CRRT (CL CRRT) and clearance via non-CRRT pathways (CL non-CRRT) to understand the relative contribution of dialysis and other factors (e.g., organ function) in eliminating antibiotics [36]. This will provide valuable data to improve individualized dosing regimens. The results from the multi-compartmental model will yield information on the time course of drug concentrations in both the central compartment (plasma) and the ultrafiltrate, allowing for the development of more accurate pharmacokinetic models.

In conclusion, the expected results from this study will lead to a better understanding of the pharmacokinetics of beta-lactam antibiotics in critically ill patients with AKI and septic shock undergoing CVVHDF, allowing for improved therapeutic strategies.

## Figures and Tables

**Table 1 antibiotics-14-00420-t001:** Guide for initial dosage flows based on weights.

Weight (kg)	Blood Flow (mL/min)	Dialysis Fluid Flow (mL/h)	Post-Filter Substitution Fluid Flow (mL/h)	Actual Effluent Dose Obtained (mL/kg/h)
50	100	1000	200	37 mL/kg/h
60	110	1100	400	37 mL/kg/h
70	120	1200	500	35 mL/kg/h
80	130	1300	500	33 mL/kg/h
90	140	1500	500	31 mL/kg/h
100	150	1400	600	31 mL/kg/h
110	160	1600	700	30 mL/kg/h
120	170	1700	800	30 mL/kg/h
130	180	1800	800	30 mL/kg/h

## Data Availability

No new data were created or analyzed in this study. Data sharing is not applicable to this article.

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
