# Peer review of "Antimicrobial Dosing During Continuous Venovenous Hemodiafiltration in Septic Shock Patients: A Prospective, Multicenter Study Protocol"

_antibiotics, 2025, doi:10.3390/antibiotics14040420_

Round 1
Reviewer 1 Report
Comments and Suggestions for Authors
First of all, I want tothank you for this great study. But I have some questions about the study design. How many cases do you plan to include in the study for each antibiotic and how did you calculate the total number of patients?
Author Response
Dear Editor, dear Reviewers,
On behalf of my team, I would like to express our sincere gratitude for your time and effort in reviewing our manuscript. Your critical insight has been key to the improvement of this work. We truly appreciate your willingness to provide valuable insights and constructive feedback. In this response letter, we have addressed your comments in detail. We remain open to any further suggestions you may have.
Response to Reviewer #1:
We thank the reviewer for his insightful question regarding the planned sample size for our study. We appreciate this feedback, which allows us to provide further clarification on our methodology.
In response to the query, we plan to include 10 patients for each betalactam antibiotic subject of study. As explicitly stated in the "Study population" section of our protocol , "Patients will be divided in 14 groups of 10 patients each (10 patients per each betalactam). The following beta lactam antibiotics will be evaluated: amoxicillin/clavulanic acid, piperacillin/tazobactam, ampicillin/sulbactam, ceftriaxone, cefotaxime, cefepime, ceftazidime/avibactam, ceftolozane/tazobactam, imipenem/cilastatin, meropenem, ertapenem, meropenem/vaborbactam, cefiderocol, aztreonam-avibactam.
These patients will be recruited from the post-surgical intensive care units (ICUs) of Hospital Clínico Universitario de Valladolid and Hospital Clínico Universitario de Valencia. They will meet the SEPSIS-3 criteria for septic shock and have acute kidney injury (AKI) requiring continuous venovenous hemodiafiltration (CVVHDF).
Regarding the absence of a traditional statistical power analysis for our sample size calculation, it is important to note that pharmacokinetic (PK) studies aimed at developing PK/PD models, such as ours, often employ different rationales for determining sample size compared to clinical trials designed to demonstrate efficacy. Our primary objective is to characterize the pharmacokinetic parameters (e.g., clearance, volume of distribution) of a range of betalactam antibiotics in the specific and understudied population of septic shock patients undergoing CVVHDF. Subsequently, we aim to use this rich pharmacokinetic data to develop robust PK/PD models that can inform optimal dosing strategies for these critical medications.
The decision to include 10 patients per antibiotic was made based on several considerations pertinent to PK/PD modeling studies:
- Parameter estimation precision: While formal power analysis for a clinical endpoint is not applicable here, including 10 patients per antibiotic group is anticipated to provide sufficient data points to obtain reasonably precise estimates of key pharmacokinetic parameters for each drug.
- PK model development: This sample size per group will allow for the application of both non-compartmental and multi-compartmental pharmacokinetic modeling techniques to describe the concentration-time profiles of the antibiotics in plasma and dialysate filtrate. As highlighted in the "Statistical analysis" section of our protocol, these analyses are central to our study .
- Monte Carlo Simulations for PK/PD Breakpoints: A crucial aspect of our study is the use of Monte Carlo simulations (with a planned 10,000 iterations as mentioned in section 3.2.2.2) to determine PK/PD breakpoints and predict the probability of target attainment for various dosing regimens against different Minimum Inhibitory Concentrations (MICs). A representative estimate of PK parameter variability, informed by data from 10 patients per antibiotic, is essential for these simulations to yield meaningful results. Studies like Akashita et al. and Gatti et al. illustrate the use of PK data and Monte Carlo simulations to optimize antibiotic dosing in critically ill patients undergoing renal replacement therapy.
While a formal power calculation focused on a clinical outcome was not performed, the planned sample size of 10 patients per antibiotic (total 140 patients) represents a pragmatic and justifiable approach for achieving the primary objectives of this pharmacokinetic study based on the available literature. For example, Akashita et al. studied biapenem PK/PD in 7 patients undergoing continuous hemodiafiltration (CHDF). Valtonen et al. included 6 patients in their study of piperacillin/tazobactam during CVVH and CVVHDF. Traunmüller et al. investigated ceftazidime pharmacokinetics in 12 patients on CVVH. The analysis by Gatti et al. reviewed data from 11 patients across six studies for ceftolozane/tazobactam during CRRT.
We have revised the manuscript accordingly and appreciate the reviewer’s valuable input in strengthening our study contribution.
Reviewer 2 Report
Comments and Suggestions for Authors
Dear Authors,
I have some concerns about your manuscript and I suggest you revise your manuscript as follows;
-What are the numbers after the keywords?
-If it is possible, add more updated data about the prevalences (lines 44-54).
-Add more content related to the beta-lactam antibiotics and highlight the information related to the sepsis.
-Paragraphs are not connected in the literature, you need to combine them in terms of meanings.
-I suggest you add a flow diagram to show selection and exclusion criteria more clearly.
-Why did you write this: "Only surgical sepsis is studied, without considering medical sepsis at any point.". What was the reason for this limitation?
-the amounts of the samples were written numerically and as words (fifty and 50), choose one of them.
-Where are your results? Do you only add expected results? Didn't you finish that study?
I think your study still needs to be finished and then you should consider publishing it.
Author Response
Dear Editor, dear Reviewers,
On behalf of my team, I would like to express our sincere gratitude for your time and effort in reviewing our manuscript. Your critical insight has been key to the improvement of this work. We truly appreciate your willingness to provide valuable insights and constructive feedback. In this response letter, we have addressed your comments in detail. We remain open to any further suggestions you may have.
Dear reviewer,
Thank you for the insightful feedback on our manuscript. We appreciate the suggestions for improvement and will address each point as follows:
Specific comments:
- "What are the numbers after the keywords?"
In response to your question regarding the numbers behind each keyword, please be informed that their inclusion was based on the formatting guidelines provided in the Antibiotics Microsoft Word template file.
Response to reviewer:
- "If it is possible, add more updated data about the prevalences (lines 44-54)."
We appreciate the reviewer's suggestion and will certainly take it into account for the revision of our study.
- "Add more content related to the beta-lactam antibiotics and highlight the information related to the sepsis."
We appreciate the reviewer's suggestion and will certainly take it into account for the revision of our study.
- "Paragraphs are not connected in the literature, you need to combine them in terms of meanings."
Response to the reviewer:
This feedback will be carefully considered during the revision process. We will work to ensure that the paragraphs are logically connected and that the narrative flows smoothly, enhancing the overall readability and understanding of the study protocol.
- "I suggest you add a flow diagram to show selection and exclusion criteria more clearly."
Response to the reviewer:
This is a valuable suggestion, that will enhance clarity regarding the study population. We will include a flow diagram in the revised manuscript to visually illustrate the patient selection process, including the eligibility criteria and any exclusion criteria that may be further specified in the full protocol.
- "Why did you write this: 'Only surgical sepsis is studied, without considering medical sepsis at any point.' What was the reason for this limitation?"
Response to the reviewer:
Thank you for your insightful question regarding the limitation of our study to surgical sepsis. In response, we would like to clarify that the decision to focus solely on surgical sepsis, without considering medical sepsis at any point, stems primarily from the characteristics of the patients we typically manage in our unit. We exceptionally treat patients who have not undergone prior surgical intervention.
Therefore, the main reason for including only postoperative patients with septic shock is that our patient population in our ICU generally consists of individuals who have undergone surgical procedures.
- "The amounts of the samples were written numerically and as words (fifty and 50), choose one of them."
Response to the reviewer:
We gladly accept the reviewer's comment and make the appropriate change.
- "Where are your results? ¿Do you only add expected results? ¿Didn't you finish that study?"
Response to the reviewer:
Thank you for your request for clarification on the absence of results in the submitted document. We appreciate your attention to this matter.
We would like to respectfully point out that the title of the document, "ANTIMICROBIALS DOSING DURING CONTINUOUS VENOVENOUS HEMODIAFILTRATION IN SEPTIC SHOCK PATIENTS: A PROSPECTIVE, MULTICENTER STUDY PROTOCOL", clearly indicates that this manuscript presents the protocol for a prospective study.
Furthermore, the section titled "2. Experimental design" outlines the planned procedures and the methods for data collection that will be employed once the study is underway.
Therefore, the current manuscript serves to detail the intended methodology and the expected data to be collected upon the study's completion. As this document is a study protocol, the results are not yet available and, consequently, are not included within this submission.
We hope this explanation clarifies the nature of the submitted document. We are looking forward to conducting the study and subsequently sharing the results in a future submission.
- "I think your study still needs to be finished and then you should consider publishing it."
Response to the reviewer:
As we have previously mentioned in response to your earlier query, the submitted manuscript, titled "ANTIMICROBIALS DOSING DURING CONTINUOUS VENOVENOUS HEMODIAFILTRATION IN SEPTIC SHOCK PATIENTS: A PROSPECTIVE, MULTICENTER STUDY PROTOCOL" , is indeed a study protocol. This document outlines the planned research, including the experimental design, the variables to be studied , and the intended procedures for data collection.
Therefore, as you rightly point out, the study needs to be conducted, the data collected and analyzed, and the results interpreted before a full research article presenting findings can be published. The current publication of this protocol serves a different purpose: to clearly delineate the study design and methodology prior to the data collection.
We appreciate your thorough review and believe that understanding the nature of this submission as a study protocol clarifies the absence of results at this stage. We appreciate your feedback and will proceed with any necessary revisions to ensure the clarity and quality of our manuscript protocol.
Reviewer 3 Report
Comments and Suggestions for Authors
The study "Antimicrobials Dosing during Continuous Venovenous Hemodiafiltration in Septic Shock Patients: A Prospective, Multicenter Study Protocol" is well-written, logically structured and presents an important contribution to the field of Kidney injury due to septic shock. Overall the article is well-written and presents compelling contribution to the field of antibiotics. The introduction effectively outlines the need of optimized protocol offering a concise background that highlight the significance of the study. Methodology is well-detailed including table illustrations.
Minor issue:
However, I assume there is a typographical error in "line 189", the unit for - 5 ampoules of calcium chloride 9.14 meEq/10ml; In my opinion the correct unit is mEq for electrolytes.
Recommendation: Correct the identified typo as specified above.
I belive the artivle will be a significant addition to the journal.
Thank You
Author Response
Dear Editor, dear Reviewers,
On behalf of my team, I would like to express our sincere gratitude for your time and effort in reviewing our manuscript. Your critical insight has been key to the improvement of this work. We truly appreciate your willingness to provide valuable insights and constructive feedback. In this response letter, we have addressed your comments in detail. We remain open to any further suggestions you may have.
Response to Reviewer #3:
Thank you for the positive and constructive feedback on our study protocol. We are pleased that you found our manuscript to be well-written, logically structured and an important contribution to the field of antibiotics optimizing prescription.
Regarding the minor issue you identified:
- You are correct in pointing out a typographical error concerning the unit for calcium chloride. We agree that the correct unit for electrolytes is mEq.
Thank you for your meticulous review and for identifying this oversight. We will proceed with the recommended correction and appreciate your time and effort in reviewing our work.
Round 2
Reviewer 2 Report
Comments and Suggestions for Authors
Thanks for your revisions.